# Ions-Induced Alginate Gelation According to Elemental Analysis and a Combinatorial Approach

**DOI:** 10.3390/ijms242216201

**Published:** 2023-11-11

**Authors:** Olga S. Zueva, Tahar Khair, Mariia A. Kazantseva, Larisa Latypova, Yuriy F. Zuev

**Affiliations:** 1Institute of Electric Power Engineering and Electronics, Kazan State Power Engineering University, 51 Krasnoselskaya Street, 420066 Kazan, Russia; ostefzueva@mail.ru (O.S.Z.); taharch937@gmail.com (T.K.); 2Kazan Institute of Biochemistry and Biophysics, FRC Kazan Scientific Center of RAS, 2/31 Lobachevsky Street, 420111 Kazan, Russia; masha353kazan@gmail.com; 3School of Applied Mathematics, HSE University, 34 Tallinskaya Street, 123458 Moscow, Russia; 4School of Chemistry and Chemical Engineering, Harbin Institute of Technology, 92 West Da-Zhi Street, Harbin 150001, China; larisa.latypova@hit.edu.cn

**Keywords:** elemental composition, alginate hydrogel, divalent metal cations, association structure, combinatorial analysis

## Abstract

This study considers the potential of elemental analysis of polysaccharide ionotropic gels in elucidating the junction zones for different divalent cations. The developed algorithm ensures the correct separation of contributions from physically adsorbed and structure-forming ionic compounds, with the obtained results scaled to alginate C_12_ block. Possible versions of chain association into dimers and their subsequent integration into flat junction zones were analyzed within the framework of the “egg-box” model. The application of combinatorial analysis made it possible to derive theoretical relations to find the probability of various types of egg-box cell occurrences for alginate chains with arbitrary monomeric units ratio μ = M/G, which makes it possible to compare experimental data for alginates of different origins. Based on literature data and obtained chemical formulas, the possible correspondence of concrete biopolymer cells to those most preferable for filling by alkaline earth cations was established. The identified features of elemental composition suggest the formation of composite hydrated complexes with the participation of transition metal cations. The possibility of quantitatively assessing ordered secondary structures formed due to the physical sorption of ions and molecules from environment, correlating with the sorption capabilities of Me^2+^ alginate, was established.

## 1. Introduction

Polysaccharides extracted from marine plants and animals (alginate, carrageenan, fucoidan, chitosan, etc.), are essential constituents of food, serving manifold functions, including the protection of users from microbial and other harmful ingredients [1]. Natural polysaccharides, in the form of solutions, hydrogels, or metal-polysaccharide microspheres [2,3,4,5,6,7], can be effective sorbents of heavy metals [8,9,10] and dangerous organic compounds [11,12,13]. Microspheres can be used not only as adsorbent but also as immobilizing mediums to preserve the biological activity of microorganisms and enzymes capable of biosorption of heavy metal ions [14,15] and dyes [16,17,18].

It should be noted that the evolution of technologies for polysaccharide gel preparation [14,15,19] has provided many abilities to modify their structure and properties by changing components and conditions of preparation, along with the use of various composite hydrogels and their integrated employment with other materials, such as proteins [5]. Accordingly, the number of fields for possible applications of these materials is sharply increasing, including the encapsulation and delivery of pharmacological agents, tissue engineering and other biomedical applications [20,21,22,23,24], creation of catalytic systems [25], removal of pesticides and antibiotics, controlled application of nutrients and plant protection products, improving soil structure, etc. [26,27].

Sodium alginate is one of the most used polysaccharides in various practical applications. This situation is representative due to its availability on the market, low cost, non-toxicity, biocompatibility, biodegradability, and ability to form ionotropic gels in the presence of divalent metal cations. First, alginates are extensively used in different food industries for many aims, e.g., thickening agents, emulsifiers, stabilizers, chelating agents, encapsulations, suspending agents, swelling agents, etc. [28,29,30]. Alginates are also characterized by some medicinal properties, in particular, antitumor, immunomodulatory [31,32] and antioxidant [33] activities, antianaphylaxis [34], anti-inflammatory [35] effects, and exert antihypertensive potential [36]. The ability of alginates to sorption of heavy metals, metalloids, fluorides, and toxic compounds is used both in medicine and for the creation of bio-sorbents and materials for water purification [8,9,10,11,12,13,37,38]. The antibacterial and wound-healing properties of alginate hydrogels, as well as the possibility of introducing drugs into their composition, are used for wound healing and the creation of wound dressing [39,40,41]. The possibility of using alginates as food additives formed the basis for the preparation of edible active food packaging films and coatings in the food packaging sector [28,42]. The invention of methods for preparing porous materials led to the design of catalytic systems [43,44] and systems for the targeted delivery of drugs, medicaments, and genes, the creation of capsules for cellular technologies, scaffolds for tissue engineering, etc. [20,21,22,23,24,45,46,47]

The large number of possible uses of biopolymers has stimulated the development of various methods for preparing hydrogels. To develop new technologies for creating composite hydrogels based on polysaccharides and new methods for improving them to obtain gel systems with desired properties, the question of the relationship between the composition and structural characteristics of the constructed hydrogels, including their dependence on the initial composition of applied polysaccharides, is of particular relevance.

To prepare hydrogel from some polysaccharides, such as alginates, they have to undergo ion-induced gelation resulting in the formation of beads or fibrils [48,49]. The structure and properties of the resulting gels and gel-based combinations noticeably depend not only on the type of polysaccharide and its original composition but also on the specific metal cation as a crosslinking agent. The mechanism of ion-induced gelation of alginates (alginic acid) using divalent ions, generally described by the “egg-box” model, was proposed 50 years ago [50]. Nonetheless, when employing various types of divalent ions, the interaction with biopolymer chains exhibits distinct characteristics, highlighting a fundamental disparity between cases involving alkaline earth and transition metals. The peculiarities of ion-polysaccharide complexation result in differences in the gel composition and microstructure [51,52,53,54,55]. In [54], using calculation methods within the framework of density functional theory, Agulhon et al. showed that the complexation between alkaline earth cations and carboxyl groups of alginate occurs only due to ionic bonds, i.e., due to electrostatic interactions. In the case of transition metal cations, according to Agulhon et al. [54], the long-range electrostatic interactions in polyelectrolyte solutions compete with stronger coordination-covalent binding of cation to alginate units. Thus, two different types of interactions lead to a general macroscopic result, which is the formation of alginate-based gel in the presence of metal cations due to the cross-linking of biopolymer chains.

In [56,57], Makarova and Zueva et al. published some aspects of the association of alginate chains into junction zones when they are bound by cations of alkaline earth and some transition metals, considering peculiar features of the elemental composition of freeze-dried hydrogel microspheres. Our theoretical considerations were based on the egg-box model of ion-induced association of polysaccharide chains proposed in [50]. In addition, we took into account that unbranched alginate chains consist of two types of structural units (β–D–mannuronic acid (M) and α–L–guluronic acid (G)), present in unequal quantities and connected in an irregular sequence. The M/G ratio depends on the alginate sample.

To explain the obtained results, we used some elements of combinatorial analysis since the formation of various structural cells is probabilistic in nature, and their filling with divalent cations depends on the type of cross-linking metal. We have shown how the data obtained can provide information on the structure of junction zones in the polysaccharide hydrogel network, the degree of filling of egg-box cells by cations, the kind and the level of cations’ interaction with alginate chains, and the most preferred types of alginate egg-box cells for cations binding. Despite the fact that for the samples studied, the M/G ratio was known as 1.56 [56], we were able to perform specific calculations only for the sample with the ratio M/G = 1.5 = 3/2 since for the combinatorial analysis, the method of compiling probability tables was used, in which the number M/G should be a simple natural fraction. The closest fraction that satisfied this condition was the fraction M/G = 1.5 = 3/2.

During the investigation of the elemental composition of ionotropic gels that formed, a notable observation emerged: in nearly all instances, the quantity of ions per segment of the polysaccharide chain exceeded what was theoretically feasible. This fact allowed us to hypothesize that some ions are present in the hydrogel structure in the form of physically adsorbed associates. The existence of metal ions in various nonequivalent positions complicates data analysis and requires the development of a correct separation algorithm for contributions from physically adsorbed and structure-forming compounds.

The purpose of this work was to generalize the method introduced by us earlier [56,57], based on combinatorial analysis, for the study of alginate gels formed by biopolymers with an arbitrary ratio μ = M/G. For this purpose, we developed an algorithm to correctly separate the contributions from cations located in nonequivalent positions, i.e., from physically adsorbed and structure-forming ionic compounds, scaling the obtained results to alginate C_12_ block. To achieve this, it was necessary to determine the types of cells that are most preferable for filling with certain cations, to evaluate their average occupation degree, and to analyze the structure of junction zones of alginate chains. In addition, one of the steps was to carry out, based on combinatorial analysis, the theoretical calculation of the probability of occurrence of egg-box cells for various alginates with an arbitrary ratio of μ = M/G, which will make it possible to compare experimental data for alginates of different origin. The obtained results make it possible to correctly determine the contribution of physically adsorbed associates and evaluate the sorption capabilities of Me^2+^ alginate. The generalization of the developed method makes it possible to use it not only for the study of alginates but also for other ionotropic gels.

## 2. Results

### Experimental Results

Dripping the sodium alginate solution into a concentrated solution of divalent metal salt made it possible to obtain microspheres of metal-alginate gel and carry out their microscopic examination. In particular, data were obtained on the morphology and elemental composition of selected zones in dried hydrogels. These data are extracted from the selected zone of the SEM image as the resulting X-ray spectrum (the dependence of X-ray counts on X-ray energy (keV)). The analysis of this spectrum is presented in the form of a table, which contains information about the type of excited lines in the X-ray spectrum, comparison standards, the relative contributions of each studied line to the total intensity, and the relative number of atoms of each element (in percent) in the sample calculated based on this data. As an example, Figure 1 shows the information obtained from the analysis of barium-alginate microspheres.

Basic information in the form of percentage elemental contribution is presented in last column of the table in Figure 1. This information enables us to precisely formulate the chemical composition of the barium alginate hydrogel as (C_51.16_O_45.81_Ba_2.74_Na_0.14_)_n_. The form is not very comparable with the standard formula for sodium alginate (C_6_H_7_O_6_Na)_n_, despite the fact that we did not include the last three elements in its composition due to their small quantities. We attribute the traces of aluminum to metal sputtering of samples for scanning electron microscopy. The presence of chlorine in such small quantities after using a concentrated BaCl_2_ solution for gelation indicates the good quality of subsequent 20 min washing after gelation.

Data on the elemental composition of studied metal-alginate hydrogels are summarized in Table 1. The analyzed hydrogels are presented in descending order of their ionic radii *r*_ion_ [52]. Two variants of strontium-alginate hydrogel microcapsules were studied: after a standard 20 min washing (line 2), during which physically adsorbed molecules remain associated with the alginate structure, and also after additional washings (line 3), during which these molecules were eliminated.

For every Me^2+^-alginate system the corresponding chemical formula can be written based on numeric data given in Table 1. In order to understand the true sense of these numerals, this formula must be modified to be compared to the chemical formula of sodium alginate (C_6_H_7_O_6_Na)_n_. Moreover, taking into account the association of alginate chains into dimers under the influence of divalent cations and the further association of dimers into flat zones of connection, it is more convenient to write this formula for a section of two monomer units. Thus, in terms of C_12_ block of two monomer units, the chemical formula of metal-alginate takes the form (C_12_H_14_O_12_MeX)_n_. Coefficient X is the average number of cations per C_12_ block introduced for convenience of comparison with the experiment. It should be noted that the complete replacement of sodium cations with Me^2+^ cations leads to the value X = 1. Any changes in the formula give a signal on the presence of some processes changing the overall result.

To transform the results of elemental analysis into a form corresponding to the C_12_ block, it is necessary to recalculate values in every line of Table 1 corresponding to a specific hydrogel to make the coefficient at C equal to 12, retaining at the same time the proportion between all elements. This can be done by multiplying all figures in the corresponding line by the factor 12/(*K*_C_), where *K*_C_ is coefficient that determines contribution of carbon to the composition of the hydrogel under study. The *K*_C_ values are written to the right of the corresponding metal. The recalculated coefficients are shown in Table 2. The contributions of Al and Si were discarded due to their smallness.

The resulting elemental composition does not include hydrogen due to its lack of characteristic X-ray emission. Its contribution must be added to the formula in proportions specific to alginates. Partially due to the peculiarities of this method, the experimentally obtained number of oxygen molecules per C_12_ block, which is slightly less than the theoretical one, may also be included.

The main feature of the given elemental composition is the difference between coefficient X (the number of cations connecting two units of the C_12_ block) and the number one. In particular, we see that for Ba^2+^-alginate, X = 0.64. For other metal-alginate gels, the coefficient X also differs from one, and in many cases, X > 1, which, at first glance, is inexplicable for flat junction zones since it exceeds the maximum theoretical value possible for the Grant model. The exception is Cu^2+^-alginate, where the indicated coefficient is almost equal to one, but the formula has another form that requires explanation. The case where X < 1 corresponds to a situation where some of the possible locations for cations remain unoccupied due to their structural features. Accordingly, in order to explain the results obtained, it is necessary to discuss possible differences in the connection of chains built from two types of structural units, as well as possible options for junction zones and their filling with cations.

## 3. Discussion

### 3.1. Junction Zone Structures

As noted earlier, alginate can be considered a natural copolymer with an irregular structure of M and G units. Each linear alginate chain, due to the structural features of monomer units, can be considered a curved chain (Figure 2a), having cavities of different depths, since M sections, in contrast to G ones, are almost flat. Schematically, it can be represented as a zigzag line (Figure 2c,d). The composition of the studied alginate is characterized by μ equal to the M/G ratio. The coefficient μ = M/G determines its structural and dynamic properties. In our experiment, we used alginate with μ = 1.56.

When divalent metal cations Me^2+^ are added, the formation of polyelectrolyte complexes occurs, depending on the type of metal ions, leading to the pairwise connection of adjacent alginate chains, i.e., the formation of alginate dimers. To describe the mechanism of cross-linking of alginate chains and their further association into flat junction zones, Grant proposed the egg-box model [50], which uses the suggested schematic representation of alginate chains in the form of zigzag lines. Although this model has been improved [58,59,60,61,62,63,64,65,66,67,68,69,70], it is still used to describe the association of alginate chains. Subsequently, it was clarified that the cross-linking of alginate chains with ions of alkaline earth metals, primarily calcium, occurs in several stages [56,63]. The first most significant step is the formation of dimers. The binding of alginate chains by divalent cations into dimers leads to appearance of a sequence of cells from 4 units (two pairwise connected blocks) with cations in the internal cavity. The spatial structures, formed by the MM, MG, and GG blocks of each alginate chain differ quite strongly. The variety of cells of dimers and the possibilities of their binding by cations also vary greatly. When sections formed from GG units of one chain and GG units of another chain are connected, a dimer is formed, the cavities of which, formed by four monomeric G units, are the most optimal binding site for any divalent cation (Figure 2b). Such cavities are often called the egg-box cells. A schematic view of the dimer with completely occupied cells is shown in Figure 2c. It should be noted here that depending on the type of cross-linking cation, this situation can be observed not only from connecting sections of alginate chains from monomeric guluronic units. Moreover, the use of transition metal cations as cross-linking agents leads, as a rule, to the formation of full-fledged dimers (Figure 2c) regardless of the type of monomer units [56]. On the contrary, alkaline earth metal cations are characterized by compounds with partially occupied cells (Figure 2d). To depict alginate chains in Figure 2c,d, the zigzag lines are used, as in the work of Grant [50]. The absence of a blue ball in the cell indicates the absence of cation. In this case, the excess charge of carboxyl groups of biopolymer chains is neutralized by hydrogen ions.

The consequence of further lateral association of dimers is the appearance of junction zones in the form of flat sheets. There are two possible ways of association of dimers, leading to three types of connection zones, clearly differing in the average occupation number (the average number of cations per C_12_ block). The zone with completely filled egg-box cells of the sheet of parallel-connected biopolymer chains, corresponding to the average occupation number X = 1 (Figure 3a), is observed for alginates of some transition metals, as shown in [56]. In this case, there is one ion Me^2+^ for every C_12_ block of two monomer units.

Association of dimers, due to van der Waals interactions and hydrogen bonds (Figure 3b), should be observed for calcium alginate. This fact was established based on X-ray diffraction data [64]. Here, on average, one cation Me^2+^ is less than for each dimer egg-box cell of four monomer units, which means that this number should not exceed 0.5 per C_12_ block. The intermediate variant (Figure 3c) corresponds to the association of dimers and biopolymer chains in the presence of electrostatic interactions of metal ions with carboxyl groups of alginate, where the formation of dimers and addition of chains occur simultaneously with the formation of junction zones. In this case, 0.5 < X < 1.0.

Thus, the use of the egg-box model and elemental composition data on the average number of Me^2+^ cations per C_12_ block provides important information about the structure formation of alginates and the type of their junction zones.

### 3.2. Application of Combinatorial Approaches to Analyze the Distribution of Cells in Connection Zones

The importance of the initial composition of the studied alginate, specifically the information on the coefficient μ = M/G, as well as the determination of sequential structure and composition, has been marked by many authors. In particular, in [71], it was noted that the monomer allocation in alginates is the result of complex biochemical occasions, the quantification and correct identification of which require relatively long monomer sequences to be known with very high accuracy. Certainly, the knowledge of the bare monomeric alginate composition does not suffice to determine the sequential structure of alginate. Some efforts to characterize the probabilistic distribution of monomer components along the biopolymer chain were made in [72,73,74].

To estimate the probability of the occurrence of possible junction zone structures, in [36], we applied an approximation using combinatorial approaches. We took into account that in the studied sodium alginate sample, the ratio μ = M/G = 1.56 can be approximated to 1.5, and we can assume that G:M = 2:3. This approach made it possible to use the method of compiling tables of variants to calculate the probabilities of block formation from two different monomeric units. The calculations carried out allowed us to interpret our experimental results quite well. However, the calculations presented in [56] were performed for a specific value of μ and therefore could not be extended to other samples. In this work, generalizing the previously used approaches, we were able to calculate probabilities of the formation of possible junction zone structures for arbitrary μ, i.e., for any alginate sample. Thus, we also clarified our previously obtained results. Table 3 presents the calculated results of probabilities for the occurrence of possible structures in alginate chain and in junction zones for arbitrary μ in analytical form, and numerical calculations were carried out for μ = 1.56 and μ = 1.5.

Table 4 presents the results of calculations for probabilities of the formation of possible structures under the connection of alginate chains in junction zones for arbitrary μ in analytical form. Also, numerical calculations were carried out for the cases μ = 1.56 and μ = 1.5. The GM-GM cells were considered equivalent to MG-MG cells but differ from cells of the GM-MG and MG-GM types.

### 3.3. Analysis of Results

#### 3.3.1. Ba^2+^-Alginate

The resulting elemental composition (Table 2) provides a basis to write the chemical formula for barium alginate in the form (C_12_H_14_O_10.75_Na_0.03_Ba_0.64_)_n_. This formula aligns well with the theoretical formula (C_12_H_14_O_12_MeX)_n_. Moreover, it was found that in this case, X = 0.64 < 1. This result indicates that the formation of junction zones (Figure 3c) corresponds to the association of alginate chains due to electrostatic interactions, where metal ions seem to fasten each dimer of the alginate chain to already formed junction zones. At the same time, some cells are inconvenient for the introduction of large barium ions, and therefore, they remain unoccupied. According to available literature data [52,53,75,76,77,78,79,80,81], barium prefers to bind to cells containing GG and MM blocks. Accordingly, GG–GG, GG–MM, MM–MM, and possibly GM-MM can be selected as the most probable Ba^2+^-containing structures. The full theoretical probability to fill these structures is equal to
X_T_ = (1 + 2μ^2^ + 4μ^3^ + μ^4^)/(1 + μ)^4^.(1)

Substituting the value μ = 1.56 lets us define the average cell occupation number as X_T_ = 0.63, which is quite close to the experimentally obtained value X = 0.64.

For further reasoning, it is also important to note that chlorine ions were almost completely removed as a result of a standard 20-min washing, i.e., they are indifferent to the formed alginate structure. It should also be noted that elemental analysis did not detect the additional presence of any other structures in the gels.

#### 3.3.2. Sr^2+^-Alginate

The elemental composition of strontium alginate (Table 2), written as C_12_O_11.6_Na_0.1_Sr_1.80_Cl_2.30_, initially contradicts the Grant model, as flat junction zones cannot have an occupation number X greater than one. The presence of chlorine and an anomalously large number of strontium ions per C_12_ block indicates the presence of structures not inherent to strontium alginate, located outside flat junction zones. Since a distinctive feature of natural polysaccharides is the presence of sorption capacity, we assumed that SrCl_2_ associates could be physically adsorbed onto resulting flat sheets of alginate chains due to the presence of local energetically favorable positions for them near biopolymer chains [56]. Calculations carried out using molecular dynamics methods [54,82] showed the existence of many sites near alginate chains suitable for binding various ions and molecules. The number of adsorbed associates per C_12_ block serves as a criterion for the sorption capacity of the studied alginate system.

Therefore, it is possible to assume the existence of strontium ions in two fundamentally different nonequivalent positions: Sr^2+^-cations, chemically bound to alginate chains, and physically adsorbed SrCl_2_ associates. The separation of contributions from strontium ions led to the following chemical formula for strontium alginate: (C_12_H_14_O_11.6_Na_0.1_Sr_0.65_ + 1.15·SrCl_2_)_n_. To test the hypothesis for the existence of physically adsorbed associates, we increased the washing time of microspheres to 6 h. This washing made it possible to completely remove associates weakly bound to the alginate structure and obtain a chemical formula very close to (C_12_H_14_O_11.33_Sr_0.65_)_n_. Both formulas indicate a cell occupation number X = 0.65.

The most preferable interactions of strontium are with GG and GM blocks [52,53,54,70,71,80,81,83]. If GG–GG, GG–GM, GG–MM, and GM–MM cells are chosen as the most probable structures containing Sr^2+^, the full theoretical probability to fill them will be equal to
X_T_ = (1 + 4μ + 2μ^2^ + 4μ^3^)/(1 + μ)^4^.(2)

Substituting μ = 1.56 value lets us define the average cell occupation number as X_T_ = 0.64, which is close to the value X = 0.65 obtained in our experiments. Despite the fact that we cannot unequivocally state that these cells will be exactly occupied due to contradictory literature data, certain considerations can still be made. The experimental value of X = 0.65 indicates that the mechanism of forming junction zones (Figure 3c) is similar to the corresponding mechanism in barium alginate, and corresponds to an association of alginate chains where metal ions seem to fasten each dimer of the alginate chain to already formed junction zones. In the case of strontium alginate, some cells also turned out to be suboptimal for the introduction of strontium ions and the formation of bonds with alginate chains, thus remaining unoccupied.

#### 3.3.3. Ca^2+^-Alginate

The elemental composition obtained in the experiment (Table 2) corresponds to the formula C_12_O_10.67_Na_0.14_Ca_1.80_Cl_3.09_. Just as in the case of strontium alginate, this formula represents contributions from calcium ions located in nonequivalent positions. Highlighting the contribution of physically adsorbed associates, the chemical formula for calcium alginate can be written as (C_12_H_14_O_10.67_Na_0.14_Ca_0.26_ + 1.54·CaCl_2_)_n_. A comparison of this formula with the previous one indicates a noticeably smaller contribution of calcium ions to the formation of bonds between alginate chains. The average occupation number of egg-box cells in this case is X = 0.26, corresponding to bonds occurring only within dimers (Figure 3b). Further lateral association of dimers when they join in junction zones occurs due to van der Waals interactions and hydrogen bonds, resulting in the low strength of such gels [75]. However, the resulting chemical formula suggests that the sorption capacity of calcium alginate is higher than for strontium alginate, and a large number of unfilled cells additionally promote the adsorption of heavy metals [10].

According to the literature, calcium ions form coordination bonds only with cells containing GG blocks, which, due to their configuration, have cavities of the largest size. There are few such cells, so the cross-linking with calcium cations leads to weak binding of alginate chains [54,83]. The probability of finding cations in the cells containing GG blocks, namely: GG–GG, GG–GM, GG–MG, and GG–MM, is
X_T_ = (1 + 4μ+ 2μ^2^)/(1 + μ)^4^.(3)

Substituting μ = 1.56 makes it possible to calculate the average cell occupation number as X_T_ = 0.28, close to the value of X = 0.26 obtained in experiment. It should be noted that the certainty of literature data and the close agreement of the found cell occupation number make it possible to unambiguously identify the cells occupied by calcium cations in Ca^2+^-alginate.

Despite the fact that in this system, the probability of the formation of the most “convenient” GG–GG cells for placing cations as a whole is quite small, as determined by the relation:X_T_ = 1/(1 + μ)^4^,(4)
(for μ = 1.56 it is approximately equal to 2.3%), their occurrence triggers the zipping mechanism of neighboring chains into dimers.

#### 3.3.4. Zn^2+^-Alginate

The elemental composition obtained in the experiment (Table 2) corresponds to the formula C_12_O_15.14_Zn_1.48_S_0.85_. Zinc sulfate was used to prepare zinc alginate hydrogels. This is associated with the appearance of physically adsorbed associates in the form of ZnSO_4_. Unlike other transition metals studied, zinc has the largest ionic radius (0.074 nm), and the Zn^2+^-induced hydrogels are similar to hydrogels of alkaline earth metals in their properties [53,54,84,85,86]. The separation of associates’ contributions led to the chemical formula (C_12_H_14_O_11.74_Zn_0.63_ + 0.8·ZnSO_4_)_n_.

The average occupation number of zinc ions per C_12_ block turned out to be X = 0.63, which approximately corresponds to barium-alginate hydrogels, and shows that association of alginate chains upon binding with zinc follows the pattern shown in Figure 3c. It is possible that zinc cations Zn^2+^ bind to the same blocks GG–GG, GG–MM, MM–MM, and GM-MM. In this case, the full theoretical probability of filling these structures is equal to
X_T_ = (1 + 2μ^2^ + 4μ^3^ + μ^4^)/(1 + μ)^4^.(5)

At μ = 1.56, the average cell occupation number is X_T_ = 0.63, which is very close to the experimentally obtained value X = 0.64.

#### 3.3.5. Cu^2+^-Alginate

The elemental composition obtained in the experiment (Table 2) corresponds to the formula C_12_O_23.05_Cu_0.99_S_0.96_. This formula contains an anomalously large number of oxygen ions, has sulfur ions, and there is an almost exact correspondence between the number of structural cells of junction zones and the number of copper ions. This composition does not indicate the presence of ordered secondary structures arising due to the physical sorption of metal ions and associates. The presented formula makes us suspect that alginate chains are bound not by copper cations, but by more complex copper-containing hydrated complexes. Taking into account such possibilities, the chemical formula can be represented with a high degree of accuracy as {C_12_H_14_O_11_(CuSO_4_·8H_2_O)}_n_. Thus, each cell of the alginate structure can contain a complex composition based on a copper cation. This conclusion is confirmed by the findings of [53], which show that when the size of the copper ion is one and a half times smaller than that of calcium, the size of the egg-box cells in Cu^2+^-alginates is always larger than in Ca^2+^-alginates. It should also be noted that calculations carried out with the density functional theory-based method in [54] showed that in the transition metal-alginates, when chains are connected, cavities are formed that can accommodate complexes, including hydration water.

The average cell occupation number by transition metal ions (in the form of a complex composition) per C_12_ block turned out to be X = 1. All possible cells are filled.

#### 3.3.6. Ni^2+^-Alginate

The elemental composition obtained in the experiment (Table 2) corresponds to the formula C_12_O_28.75_Ni_2.99_S_2.27_. Also noteworthy here is the anomalously large number of oxygen ions, indicating the existence of hydration water, the presence of sulfur ions, and the number of nickel ions, which exceeds the number of structural cells of junction zones by exactly three times. A careful analysis of this formula leads to the conclusion that there are at least three nonequivalent positions in nickel ions, as well as its possible connections with hydration water. We assume that outside the flat sheets of junction zones, NiSO_4_·4H_2_O associates are adsorbed on them on both sides. Ni^2+^-alginate cells contain the cross-linking nickel ions. There are exactly as many of them as there are cells, i.e., X = 1. However, in this situation, there are still 0.27 sulfur ions per cell. This amount of sulfur ions is too noticeable to be attributed to experimental errors. We have to conclude that some of the cells may contain not just individual nickel ions, but complexes based on nickel ions, possibly even containing hydration water. Most likely, these complexes occupy cells with cavities of the largest size, i.e., the cells that contain GG blocks. Their number was found for the case of Ca^2+^-alginate:X_T_ = (1 + 4μ+ 2μ^2^)/(1 + μ)^4^.(6)

Numerical calculations carried out for the value μ = 1.56 gave the result X_T_ = 0.28, very close to the value of X = 0.27 obtained in the experiment.

All of the above gives reason to assume the following two composition options for Ni^2+^-alginate:

(1) {0.73·C_12_H_14_O_11.67_Ni + 0.27·C_12_H_14_O_11.67_·(NiSO_4_) + 2·(NiSO_4_·4H_2_O)}_n_;

(2) {0.73·C_12_H_14_O_10.6_Ni + 0.27·C_12_H_14_O_10.6_·(NiSO_4_·4H_2_O) + 2·(NiSO_4_·4H_2_O)}_n_.

These compositions indicates that there are nickel ions in the flat junction zones in all cells. In 73% of the cells there are single Ni ions, and in 27% of the cells that include GG regions and, accordingly, have larger cavities, these ions are in the form of NiSO_4_ complexes, possibly containing hydrate water. Since, according to elemental analysis, the resulting associates most likely contain hydration water; in other words, since such structures are energetically more favorable, we assumed that in the process of complexation of nickel cations with alginates (i.e., in the cells of junction zones), hydration water can also be involved. However, this assumption requires additional verification.

As noted earlier, associates NiSO_4_·4H_2_O are adsorbed on flat junction zones on both sides.

#### 3.3.7. Mn^2+^-Alginate

The elemental composition obtained in the experiment (Table 2) corresponds to the formula C_12_O_6.55_Na_0.03_Mn_3.09_Cl_4.51_. This composition indicates the presence of physically adsorbed associates in the form of MnCl_2_. By highlighting their contribution, one can obtain a chemical formula in the form (C_12_O_6.55_H_14_Mn + 2.1·MnCl_2_)_n_, which indicates the complete filling of all possible cells with manganese cations, i.e., X = 1. A noticeably smaller number of oxygen ions and the lacy walls of the formed hydrogel [56] give reason to assume the occurrence of processes leading to partial dehydrogenation, decarboxylation, and degradation of alginate chains to oligomers [51,87]. The instability of manganese alginates was also noted in [53,88]. The breaks in alginate chains likely result in increase in the surface area available for the physical adsorption of associates, as evidenced by the coefficient of 2.1, indicating that the number of associates per C_12_ block is slightly higher than the theoretical value 2.

## 4. Materials and Methods

### 4.1. Materials

To prepare hydrogels, we used the natural anionic polysaccharide sodium alginate (A-2033 “Sigma”, St. Louis, MO, USA, CAS registry no. 9005-38-3). Alginate consists of linear alginic acid molecules, built from two types of structural units. M units are β-D-mannuronic acid residues, and G units are α-L-guluronic acid residues [89,90,91]. In fact, alginate is a natural copolymer with an irregular block structure composed of M and G units, with the number of M units usually prevailing over the number of G units. Typically, the M/G ratio is approximately 1.5–2. The alginate used in our work was reported to have a mannuronic acid to guluronic acid ratio (M/G) of 1.56 [92,93].

Alginic acid is described by the chemical formula (C_6_H_8_O_6_)_n_, which is the same for both M units and G units. However, spatial structures formed by the MM, MG, and GG blocks differ quite significantly. The relative M/G composition and the order of alternation of uronic acids in alginates from different algae sources depend on many factors.

To prepare metal-alginate microspheres, we used salts of divalent metals (Ba, Ca, Sr, Mn, Cu, Zn, Ni, “Tatkhimproduct”, Kazan, Russia): calcium chloride (CAS registry no. 10043-52-4), barium chloride (CAS registry no. 10361-37-2), strontium chloride (CAS registry no. 10476-85-4), manganese dichloride (CAS registry no. 7773-01-5), nickel sulfate (CAS registry no. 7786-81-4), copper sulfate (CAS registry no. 7758-98-7), and zinc sulfate (CAS registry no. 7733-02-0). To prepare solutions, we used ultrapure water purified with the “Arium mini” water purification system (Sartorius, Göttingen, Germany). An aqueous solution of sodium alginate (2 wt.%) was prepared according to standard methods [16,56,94,95]. When a hot (70 °C) solution of sodium alginate was dripped into a concentrated solution of divalent metal salt (1 M) at a temperature of 20–23 °C, microspheres of composite hydrogels with a diameter of about 2 mm were instantly formed, in which monovalent sodium ions were replaced by divalent metal ions. The resulting microspheres were kept in solution for 20 min, then washed twice and frozen in liquid nitrogen for freeze-drying. The microsphere preparation procedure and washing time were the same for all samples.

### 4.2. Methods

Scanning electron microscopy and energy dispersive X-ray spectroscopy were chosen as the main experimental methods in this work. The field emission scanning electron microscope “Merlin” (“Carl Zeiss”, Jena, Germany) was used to study the microstructure and elemental analysis of freeze-dried samples of ionotropic gels prepared with sodium alginate and divalent metals Ba, Ca, Cu, Zn, Ni, and Mn. The analytical capabilities of the device were expanded by additional attachments of X-ray microanalysis Oxford Instruments INCAx-act and an electron backscatter diffraction (EBSD) recording system Oxford Instruments CHANNEL5. The studies were carried out at the Interdisciplinary Center “Analytical Microscopy” (Kazan Federal University, Kazan, Russia). The microstructure and elemental analysis of samples with Sr were investigated on the Auriga Crossbeam Workstation (Carl Zeiss AG, Jena, Germany), equipped with INCA X-Max silicon drift detector for energy dispersive X-ray microanalysis (Oxford Instruments, Abingdon, OX, UK). This research was carried out at the Shared Research Center “Applied Nanotechnology” (Kazan National Research Technical University, Kazan, Russia).

Energy dispersive X-ray spectroscopy makes it possible to study the elemental composition of certain zones of solid matter. This method is based on studying the energy distribution of X-ray spectra emitted as a result of the action of the electron beam on the atoms in the area under study. By examining the energy spectrum of such radiation, it is possible to draw conclusions about the qualitative and quantitative elemental composition of the sample, in our case, the surface zones or internal structures of freeze-dried alginate microspheres. It should be noted that this method has limitations connected to the lack of characteristic X-ray emission from hydrogen and lower accuracy in defining the quantitative contexture of slight elements like oxygen and carbon.

The information obtained, first of all, will make it possible to determine the real average number of cations X cross-linking two units of alginate chains and its differences from the theoretical value X = 1, corresponding to completely cross-linked chains.

## 5. Conclusions

This work examines the possibility of using elemental analysis data of freeze-dried polysaccharide ionotropic gels to elucidate the detailed structure of alginate gels, namely the construction of junction zones of biopolymer chains and its dependence on the type of cross-linking divalent cations. It was previously established that not all metal cations may be the cross-linking ones. In relation to polysaccharide chains, they may be in nonequivalent positions, which complicates the analysis of elemental composition. We have developed an algorithm for the correct separation of contributions from physically adsorbed compounds and transforming the results of elemental analysis into a form related to the alginate C_12_ block. Its use made it possible to recalculate the percentage of elements existing in the sample in relation to polysaccharide structural element for which the chemical formula of ionotropic gel is written, and to compare experimental and theoretical compositions. To analyze the data obtained, we also carried out a theoretical study of the emergence of possible structures when alginate chains, formed by two types of structural units (M and G), are connected into dimer with their subsequent association into junction zones. It has been established that two types of dimers, differing in the degree of cells filling by ions, lead to three types of junction zones. The use of combinatorial analysis made it possible to derive relations for finding the probability of the occurrence of egg-box cells of various types that appear during the association of alginate chains with an arbitrary link ratio μ = M/G and calculate them for the actual value μ = 1.56, which made it possible to compare with experimental data.

A consequence of the relatively weak electrostatic interaction of alkaline earth cations with alginate units is an unequal degree of their binding to different cells of the egg-box structure, which manifests itself in the presence of certain number of unoccupied spaces in flat junction zones and leads to an average cell occupation number X < 1. Based on literature data and obtained calculation formulas, a partial correspondence of the cells most preferable for filling with definite cations with certain structures was established. Despite the fact that under the cross-linking with transition metals (except zinc) the average occupation number of cells is X = 1, other features of the chemical formula written on the basis of elemental analysis made it possible to suggest the probable binding of alginate chains, not only by single metal cations but by more complex hydrated complexes based on metal ions. In particular, a similar situation, according to our experimental data, occurs in Cu^2+^ and Ni^2+^ alginates.

The existence of unequal binding sites for metal ions with alginate chains can lead to the appearance of ordered secondary structures due to the physical sorption of ions and molecules from the environment. The developed algorithm for processing the existing results of elemental analysis takes into account the possibility of correctly identifying the contribution of physically adsorbed associates that correlate with the sorption capabilities of Me^2+^-alginate. The generalization of the developed method will make it possible to use it to study not only alginate but also other ionotropic gels.

## Figures and Tables

**Figure 1 ijms-24-16201-f001:**
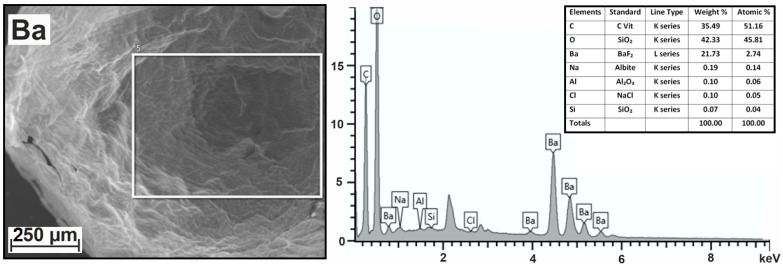
SEM image of selected area, resulting X-ray spectrum, and table containing information about the relative percentages of atoms of each element in the sample.

**Figure 2 ijms-24-16201-f002:**
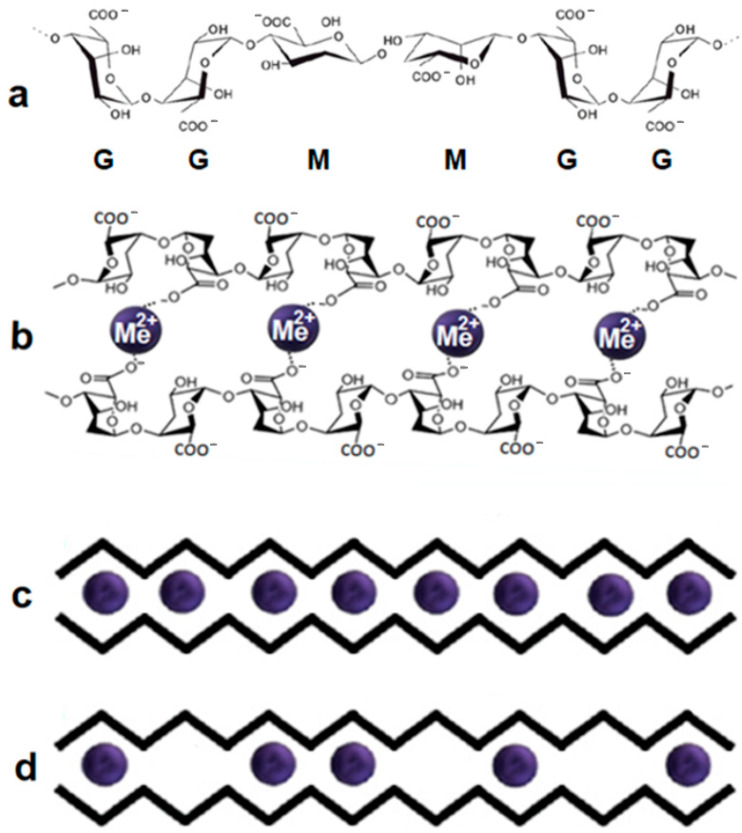
Components of alginate chains: (**a**) dimers formed at their association (**b**), schematic representation of dimer with fully (**c**), partially (**d**), and occupied cells.

**Figure 3 ijms-24-16201-f003:**
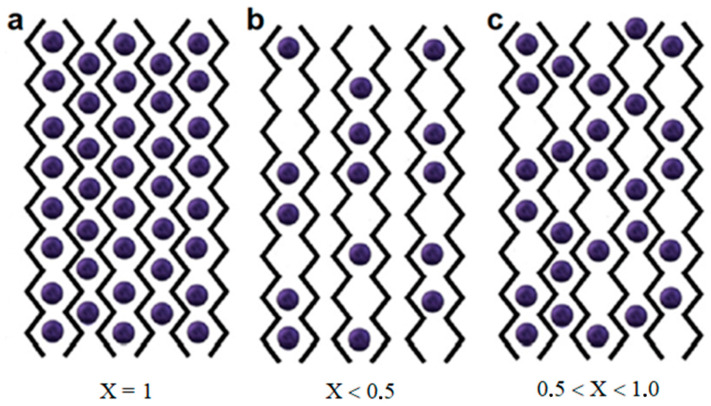
Three types of junction zones during association of dimers, differing in average occupation number X: (**a**) zone with completely filled cells (X = 1); (**b**) association of dimers due to van der Waals interactions and hydrogen bonds (X < 0.5); (**c**) association of dimers in the presence of electrostatic interactions 0.5 < X < 1.0.

**Table 1 ijms-24-16201-t001:** Elemental composition of studied hydrogels according to X-ray spectrums in atomic%.

N	*r*_ion,_ nm	Hydrogel	Me	C	O	Na	Cl	S	Al	Si
1	0.135	Ba^2+^-alginate	Ba, 2.74	51.16	45.81	0.14	0.05	–	0.06	0.04
2	0.113	Sr^2+^-alginate	Sr, 6.47	43.18	41.73	0.36	8.27	–	–	–
3	0.113	Sr^2+^-alginate (add. washed)	Sr, 2.71	50.04	47.25	–	–	–	–	–
4	0.099	Ca^2+^-alginate	Ca, 6.48	43.30	38.51	0.49	11.15	–	0.04	0.03
5	0.074	Zn^2+^-alginate	Zn, 5.02	40.68	51.33	–	–	2.87	0.05	0.05
6	0.073	Cu^2+^-alginate	Cu, 2.68	32.40	62.23	–	–	2.60	0.05	0.04
7	0.069	Ni^2+^-alginate	Ni, 6.49	26.07	62.45	–	–	4.94	0.03	0.02
8	0.067	Mn^2+^-alginate	Mn, 11.8	45.80	24.99	0.12	17.2	–	0.05	0.04

**Table 2 ijms-24-16201-t002:** Elemental composition of studied hydrogels calculated in terms of the C_12_ block.

N	Hydrogel	Me	C	O	Na	Cl	S	Elemental Composition
1	Ba^2+^-alginate	Ba, 0.64	12	10.75	0.03	–	–	C_12_O_10.75_Na_0.03_Ba_0.64_
2	Sr^2+^-alginate	Sr, 1.80	12	11.60	0.10	2.30	–	C_12_O_11.6_Na_0.1_Sr_1.80_Cl_2.30_
3	Sr^2+^-alginate (add. washed)	Sr, 0.65	12	11.33	–	–	–	C_12_O_11.33_Sr_0.65_
4	Ca^2+^-alginate	Ca, 1.80	12	10.67	0.14	3.09	–	C_12_O_10.67_Na_0.14_Ca_1.80_Cl_3.09_
5	Zn^2+^-alginate	Zn, 1.48	12	15.14	–	–	0.85	C_12_O_15.14_Zn_1.48_S_0.85_
6	Cu^2+^-alginate	Cu,0.99	12	23.05	–	–	0.96	C_12_O_23.05_Cu_0.99_S_0.96_
7	Ni^2+^-alginate	Ni, 2.99	12	28.75	–	–	2.27	C_12_O_28.75_Ni_2.99_S_2.27_
8	Mn^2+^-alginate	Mn, 3.09	12	6.55	0.03	4.51	–	C_12_O_6.55_Na_0.03_Mn_3.09_Cl_4.51_

**Table 3 ijms-24-16201-t003:** Probabilities of possible structures when alginate chains form dimers.

N	Structure	Probability (Arbitrary μ)	Probability (μ = 1.56)	Probability (μ = 1.5)
1	GG	1/(1 + μ)^2^	0.153	0.16
2	GM	μ/(1 + μ)^2^	0.238	0.24
3	MG	μ/(1 + μ)^2^	0.238	0.24
4	MM	μ^2^/(1 + μ)^2^	0.371	0.36

**Table 4 ijms-24-16201-t004:** Probabilities of possible structures in junction zones.

N	Structure	Probability (Arbitrary μ = M/G)	Probability (μ = 1.56)	Probability (μ = 1.5)
1	GG–GG	1/(1 + μ)^4^	0.0233	0.0256
2	GG–GM, GG–MG	4μ/(1 + μ)^4^	0.1453	0.1536
3	GG-MM	2μ^2^/(1 + μ)^4^	0.1133	0.1152
4	GM–GM	2μ^2^/(1 + μ)^4^	0.1133	0.1152
5	GM-MG	2μ^2^/(1 + μ)^4^	0.1133	0.1152
6	MM–GM, MM–MG	4μ^3^/(1 + μ)^4^	0.3536	0.3456
7	MM–MM	μ^4^/(1 + μ)^4^	0.1379	0.1296

## Data Availability

Data are contained within the article.

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
