# Peer review of "Ions-Induced Alginate Gelation According to Elemental Analysis and a Combinatorial Approach"

_ijms, 2023, doi:10.3390/ijms242216201_

Round 1

Reviewer 1 Report

Comments and Suggestions for Authors

Author proposed a paper entitled “Ions-Induced Alginate Gelation According to Elemental Analysis and Combinatorial Approach” for the publication in IJMS.

The paper has a good scientific soundness.

I suggest adding an abbreviation list, according to the guidelines of this Journla.

Here is the list of my issues:

Line 52. I would rewrite this “However, for each type of diva lent ions used, the complexation with biopolymer chains has its own features, which are of fundamental difference for the cases of alkaline earth and transition metals” as follows: “Nonetheless, when employing various types of divalent ions, the interaction with biopolymer chains exhibits distinct characteristics, highlighting a fundamental disparity between the cases involving alkaline earth and transition metals.”

Line 56. “In [34],” I would cite the name of the first author of this paper such as “X et al.”

Line 64. “In [36,37]” same as above.

Line 84. Concerning this sentence “In the course of studying the elemental composition of formed ionotropic gels, it 84 turned out that in almost all cases the number of ions per section of polysaccharide chain 85 is greater than the theoretically possible “, I suggest rewriting it as: “During the investigation of the elemental composition of ionotropic gels that were formed, a notable observation emerged: in nearly all instances, the quantity of ions per segment of the polysaccharide chain exceeded what was theoretically feasible.”

Line 91. “by us early method []” was this a missing reference?

Material section needs to be enriched with CAS numbers of all the components.

I suggest separating table of elements and weights by the X-ray spectrum. Please provide a table 1 with quantitative information.

Line 173. The sentence has some mistake: “These data allow us to write the exact chemical formula of barium alginate hydrogel these data allow us to write the exact chemical formula of barium alginate hydrogel”. It could be changed as follows: “This information enables us to precisely formulate the chemical composition of the barium alginate hydrogel.”

Line 214. “X (the number of cations connecting two units of C12 block)” I suggest moving this explanation in Methods section, where variables could be generally defined.

Line 237. “When divalent metal cations Me2+ are added the formation of polyelectrolyte com-237 plexes depending on the type of metal ions occurs, leading to pairwise connection of ad-238 jacent alginate chains, i.e. to formation of alginate dimers.” Please check the syntax in this period.

Line 353. “So, we assumed the existence” please, use more impersonal forms such as “therefore, it is possible to assume…”

Line 426 “The presented formula makes us to suspect” similar than previous comment.

Comments on the Quality of English Language

The use of English and the syntax needs to be revised in the manuscript.

Author Response

Reviewer #1:

Author proposed a paper entitled “Ions-Induced Alginate Gelation According to Elemental Analysis and Combinatorial Approach” for the publication in IJMS.

The paper has a good scientific soundness.

I suggest adding an abbreviation list, according to the guidelines of this Journal.

Here is the list of my issues:

Answers:

  1. Line 52. I would rewrite this “However, for each type of divalent ions used, the complexation with biopolymer chains has its own features, which are of fundamental difference for the cases of alkaline earth and transition metals” as follows: “Nonetheless, when employing various types of divalent ions, the interaction with biopolymer chains exhibits distinct characteristics, highlighting a fundamental disparity between the cases involving alkaline earth and transition metals.”

Answer: We rewrote this sentence.

  1. Line 56. “In [34],” I would cite the name of the first author of this paper such as “X et al.”

Answer: We rewrote this sentence and introduced the name of the first author of this paper as “Agulhon et al.”

  1. Line 64. “In [36,37]” same as above.

Answer: We rewrote this sentence and introduced the names of the first authors of this paper as “Makarova and Zueva et al.”

  1. Line 84. Concerning this sentence “In the course of studying the elemental composition of formed ionotropic gels, it 84 turned out that in almost all cases the number of ions per section of polysaccharide chain 85 is greater than the theoretically possible “, I suggest rewriting it as: “During the investigation of the elemental composition of ionotropic gels that were formed, a notable observation emerged: in nearly all instances, the quantity of ions per segment of the polysaccharide chain exceeded what was theoretically feasible.”

Answer: We rewrote this sentence according with comments.

  1. Line 91. “by us early method []” was this a missing reference?

Answer: We added the missing reference.

  1. Material section needs to be enriched with CAS numbers of all the components.

Answer: We added the CAS numbers of all the components.

  1. I suggest separating table of elements and weights by the X-ray spectrum. Please provide a table 1 with quantitative information.

Answer: In our Table 1 the elemental composition of studied hydrogels is separated according to X-ray spectra in atomic %. We have added words “according to X-ray spectrums” to the title of the table.

  1. Line 173. The sentence has some mistake: “These data allow us to write the exact chemical formula of barium alginate hydrogel these data allow us to write the exact chemical formula of barium alginate hydrogel”. It could be changed as follows: “This information enables us to precisely formulate the chemical composition of the barium alginate hydrogel.”

Answer: We rewrote this sentence according to comments.

  1. Line 214. “X (the number of cations connecting two units of C12 block)” I suggest moving this explanation in Methods section, where variables could be generally defined. X (количество катионов, соединяющих две единицы блока C12)»

Answer: We added additional information to section Methods: The information obtained, first of all, will make it possible to determine the real average number of cations X cross-linking two units of alginate chains and its differences from the theoretical value X = 1, corresponding to completely cross-linked chains.

  1. Line 237. “When divalent metal cations Me2+ are added the formation of polyelectrolyte com-237 plexes depending on the type of metal ions occurs, leading to pairwise connection of ad-238 jacent alginate chains, i.e., to formation of alginate dimers.” Please check the syntax in this period.

Answer: We checked the syntax: When divalent metal cations Me2+ are added, the formation of polyelectrolyte complexes occurs depending on the type of metal ions, leading to the pairwise connection of adjacent alginate chains, i.e., to formation of alginate dimers.

  1. Line 353. “So, we assumed the existence” please, use more impersonal forms such as “therefore, it is possible to assume…”

Answer: We rewrote this sentence.

  1. Line 426 “The presented formula makes us to suspect” similar than previous comment.

Answer: We rewrote this sentence. “The presented formula makes us to suspect  indicates the possibility the existence of strontium ions in two fundamentally different nonequivalent positions”

  1. The use of English and the syntax needs to be revised in the manuscript.

Answer: We checked and improved carefully our English using the assistance of native holder

Dear Reviewer, we thank you for your time spent for our manuscript and your useful comments which we tried to take into account in revised version.

Reviewer 2 Report

Comments and Suggestions for Authors

Dear the Editor

Zueva OS et al reported that mechanism of egg-box-typed gelation of alginate hydrogel induced by a variety of divalent metal cations best described with Ca2+. Data were obtained by X-ray elemental analysis. These authors first introduced the m value defined as the ratio of M to G (Fig. 2). Then, based on this value, four possible structures of alginate chain polymer (ie GG, GM, MG, and MM) and 6 junction zone structures were classified (Tables 3 and 4). Results were descriptively presented. Due to a limited discussion for biological application, this manuscript appeared to be more suitable for the Polymers journal.

Major concerns:

1)     For the readers of IJMS, some biological application should be provided. For example, the gelation of alginate is widely used in food chemistry and pharmaceutical formulation. A potential link between such biological application and the aim of this work was not clearly described. Please describe higher solubility of alginate hydrogels made by other divalent metal cations other than Ca2+, so that readers could understand more easily why these authors studied these divalent metal cations so extensively.

Minor concerns:

1)English needs to be corrected by a professional editor.

2)Text needs to be prepared with much intensive precaution (ie L91, L111 glucuronic acid, etc).

3)In LL183-185, line 2 and line 3 seemed obscure.

Comments on the Quality of English Language

English needs to be corrected by a professional editor.

Author Response

Reviewer #2

Zueva OS et al reported that mechanism of egg-box-typed gelation of alginate hydrogel induced by a variety of divalent metal cations best described with Ca2+. Data were obtained by X-ray elemental analysis. These authors first introduced the m value defined as the ratio of M to G (Fig. 2). Then, based on this value, four possible structures of alginate chain polymer (ie GG, GM, MG, and MM) and 6 junction zone structures were classified (Tables 3 and 4). Results were descriptively presented. Due to a limited discussion for biological application, this manuscript appeared to be more suitable for the Polymers journal.

Major concerns:

  1. For the readers of IJMS, some biological application should be provided. For example, the gelation of alginate is widely used in food chemistry and pharmaceutical formulation. A potential link between such biological application and the aim of this work was not clearly described. Please describe higher solubility of alginate hydrogels made by other divalent metal cations other than Ca2+, so that readers could understand more easily why these authors studied these divalent metal cations so extensively.

Answer: We have added some additional information to Introduction. We described in more detail the biological applications of alginate gelation. We added an additional 20 references 28-47. We also introduced text describing the need to consider different gel options to obtain gel systems with the desired properties:

Sodium alginate is one of the most used polysaccharides in various practical applications. This situation is representative due to its availability on the market, low cost, non-toxicity, biocompatibility, biodegradability and the ability to form ionotropic gels in the presence of divalent metal cations. First, alginates are extensively used in different food industries for many aims, e.g., as thickening agents, emulsifiers, stabilizers, chelating agents, encapsulations, suspending agents, swelling agents, etc. [28-30]. Alginates are also characterized by some medicinal properties, in particular, antitumor, immunomodulatory [31,32] and antioxidant [33] activities, antianaphylaxis [34] and anti-inflammatory [35] effects and exert antihypertensive potential [36]. The ability of alginates to sorption of heavy metals, metalloids, fluorides and toxic compounds is used both in medicine and for the creation of bio sorbents and materials for water purification [8-13,37,38]. The antibacterial and wound-healing properties of alginate hydrogels, as well as the possibility of introducing drugs into their composition, are used for wound healing and the creation of wound dressing [39-41]. The possibility of using alginates as food additives formed the basis for the preparation of edible active food packaging films and coatings in the food packaging sector [28,42]. The invention of methods for preparing porous materials led to designs of catalytic systems [43,44] and systems for targeted delivery of drugs, medicaments, and genes, the creation of capsules for cellular technologies, scaffolds for tissue engineering, etc.[20-24,45-47]

The large number of possible uses of biopolymers has stimulated the development of various methods for preparing hydrogels. To develop new technologies for creating composite hydrogels based on polysaccharides and methods for improving them to obtain gel systems with the desired properties, the question of finding the relationship between the composition and structural characteristics of the constructed hydrogels, including their dependence on the initial composition of applied polysaccharides, is of particular relevance.

Minor concerns:

1) English needs to be corrected by a professional editor.

Answer: We checked and improved carefully our English with assistance of its native holder.

2) Text needs to be prepared with much intensive precaution (ie L91, L111 glucuronic acid, etc).

Answer: We added the missing reference in L91 to []. In L111 we have written ‘’ α–L–guluronic acid’’.

3) In LL183-185, line 2 and line 3 seemed obscure.

We added the text:

Two variants of strontium alginate hydrogel microcapsules were studied: after a standard 20-minute washing (line 2), during which physically adsorbed molecules remain associated with the alginate structure, and also after additional washings (line 3), during which these molecules were eliminated.

Dear Reviewer, we thank you for your time spent for our manuscript and your useful comments which we tried to take into account in its revised version.

Round 2

Reviewer 1 Report

Comments and Suggestions for Authors

Authors provided a new version of their manuscript.

Moreover, they answered point by point to each of my raised issues.

New paragraphs were added in to the introduction section, giving more explanations and suggesting more points of discussion.

Missing references were added and the format of references was changed accordingly.

I only suggest improving the focus of table inside Figure 1b.

Reviewer 2 Report

Comments and Suggestions for Authors

Dear the Editor

All concerns raised by this Reviewer have been properly corrected.